# Y-Box-Binding Proteins Have a Dual Impact on Cellular Translation

**DOI:** 10.3390/ijms25031736

**Published:** 2024-02-01

**Authors:** Irina A. Eliseeva, Andrey I. Buyan, Egor A. Smolin, Karina S. Kaliadzenka, Sergey Popov, Ivan V. Kulakovskiy, Dmitry N. Lyabin

**Affiliations:** 1Institute of Protein Research, Russian Academy of Sciences, Pushchino 142290, Russia; yeliseeva@vega.protres.ru (I.A.E.); andreybuyanchik@gmail.com (A.I.B.); smolinegoralexeyevich@gmail.com (E.A.S.); kalia.dzenka@yandex.ru (K.S.K.); ivan.kulakovskiy@gmail.com (I.V.K.); 2Endocrinology Research Center, Moscow 117036, Russia; swpopov73@gmail.com

**Keywords:** YB-1, YB-3, translation, ribo-seq, RNA-seq, PAR-CLIP

## Abstract

Y-box-binding proteins (YB proteins) are multifunctional DNA- and RNA-binding proteins that play an important role in the regulation of gene expression. The high homology of their cold shock domains and the similarity between their long, unstructured C-terminal domains suggest that Y-box-binding proteins may have similar functions in a cell. Here, we consider the functional interchangeability of the somatic YB proteins YB-1 and YB-3. RNA-seq and Ribo-seq are used to track changes in the mRNA abundance or mRNA translation in HEK293T cells solely expressing YB-1, YB-3, or neither of them. We show that YB proteins have a dual effect on translation. Although the expression of YB proteins stimulates global translation, YB-1 and YB-3 inhibit the translation of their direct CLIP-identified mRNA targets. The impact of YB-1 and YB-3 on the translation of their mRNA targets is similar, which suggests that they can substitute each other in inhibiting the translation of their mRNA targets in HEK293T cells.

## 1. Introduction

The involvement of Y-box-binding proteins (YB proteins) in the regulation of gene expression is beyond a doubt. There is ample evidence for their participation in the control of translation, transcription, and regulation of mRNA stability (see reviews [1,2,3]). The role of YB proteins, primarily YB-1, in the development of cancer and the pathologies associated with inflammatory processes is being intensively studied (see reviews [4,5,6]). However, the full range of targets controlled by YB proteins is not currently known, and it is also unclear to what extent the functions of YB proteins overlap. The existence of common functions of YB proteins is assumed based on the almost complete identity of their cold shock domains and the presence of similar arginine-rich motifs, which are believed to determine the specificity of the interaction of YB proteins with nucleic acids [1,2,7,8]. In somatic cells, there are only two YB proteins, YB-1 and YB-3 [9]. Our previous studies showed the possibility of their interchangeability in performing their functions in the cell [10]. Specifically, with the YB-1 gene knocked out, only minor changes occur in the cell translatome and transcriptome, which likely results from a significantly increased amount of YB-3 and its ability to partially replace the level of YB-1 activity in the cell. Importantly, the sets of mRNAs that can bind to YB-1 and YB-3 are very similar. This suggests that the increased synthesis of YB-3 in the absence of YB-1 leads to the interaction of YB-3 with previously YB-1-bound mRNAs, thereby not only restoring their translation but also the translation inhibition, similar to the effect of elevated YB-1 expression [10]. Therefore, in experimental studies of the YB-1 effect on gene expression in the cell, it is necessary to keep in mind the concurrent YB-3 activity and use cells lacking both YB-1 and YB-3 to neutralize the compensatory effect of the latter.

This study demonstrates that YB proteins are functionally interchangeable at the translation level in HEK293T cells. The expression of either YB-1 or YB-3 partially restores translation previously inhibited in YB protein-null cells. However, both YB-1 and YB-3 inhibit the translation of their direct CLIP-identified mRNA targets, which encode proteins involved in translation, oxidative phosphorylation, and mitochondrion organization. 

## 2. Results

### 2.1. YB-1 and YB-3 Expressions Partially Restore Translation in HEK293TΔYB-1ΔYB-3 Cells 

As we showed previously, *YB-1* (*YBX1*) knockout induces the overexpression of YB-3, which, in turn, binds a repertoire of mRNAs similar to that of YB-1 and, likely, functionally substitutes YB-1 in translation regulation [10]. However, we did not completely rule out the possibility that adaptation factors other than YB-3 (including other RNA-binding proteins or signaling pathway activation) compensate for the YB-1 loss in translation. Here, to demonstrate the interchangeability of YB-1 and YB-3, we directly compare translational changes occurring in cells expressing one of the YB proteins (YB-1 or YB-3) with those in cells expressing neither YB-1 nor YB-3.

First, we obtained derivatives of the double knockout HEK293TΔYB-1ΔYB-3 (ΔΔ) cells stably expressing full-length (with the untranslated region) *YB-1* and/or *YB-3* mRNA. It could be expected that in such cells with *YB-1* mRNA and *YB-3* mRNA containing native UTRs, the proteins would be synthesized in quantities comparable to the amount of endogenous proteins in wild-type HEK293T since the translation of *YB-1* mRNA and *YB-3* mRNA is controlled by the YB-1 protein, as we described previously [10,11]. Indeed, the immunoblot analysis of cell lysates showed that the YB-1 protein is expressed in HEK293TΔΔ + YB-1 (ΔΔ + YB-1) and HEK293TΔΔ + YB-1 + YB-3 cells (ΔΔ + YB-1 + YB-3) at the same level as in parental HEK293T cells (from which HEK293TΔΔ was derived). It is important that in HEK293TΔΔ + YB-3 (ΔΔ + YB-3), the amount of YB-3 is noticeably higher than in all the other lines. This is explained by the fact that these cells lack YB-1, which was previously shown to be an inhibitor of *YB-3* mRNA translation. Therefore, in such cells, YB-3 synthesis is not controlled. In YB-1-expressing cells like ΔΔ + YB-1 + YB-3, the amount of YB-3 noticeably decreases to approximately the same level as in the original HEK293T cells (Figure 1a).

Next, we analyzed the global translation of the studied cells (Figure 1b). The translation in the ∆∆ cell line noticeably decreases as compared to the parent HEK293T line (the polysomal fraction is less pronounced while the monosome fraction (80S) is more pronounced). The cells expressing YB-1 and YB-3 partially restore the translation. Of interest, the expression of YB-3 has a weaker effect on the global translation than the expression of YB-1 (cf. purple and green lines, Figure 1b). We speculate that in the absence of YB-1, the expression of *YBX3* is not controlled, thus causing the increased synthesis of YB-3. In large amounts, YB-3 acts as a translation inhibitor. To validate our findings, we measured global translation by estimating azidohomoalanine incorporation into newly synthesized proteins (Appendix A). Similarly to the effect demonstrated by the gradient profiles, the global translation strongly declined in ∆∆ cells, as compared to the wild-type cells, and YB-1 or YB-3 expression partially restored the translation level. 

### 2.2. Translational Changes Correlate in YB-1- and YB-3-Expressing Cells 

In order to clarify at which step YB-1 and YB-3 could functionally substitute each other, we performed RNA-seq and Ribo-seq in double knockout HEK293TΔYB-1ΔYB-3 (ΔΔ) cells and YB-1- (ΔΔ + YB-1) or YB-3-expressing (ΔΔ + YB-3) cells.

First, we compared (Figure 2a) changes in RNA abundance (RNA-seq), ribosome footprint counts (Ribo-seq), and ribosome occupancy (RO) occurring in ΔΔ + YB-1 cells against ΔΔ + YB-3 cells (both cell lines compared to ΔΔ cells). All comparisons demonstrate a significant positive correlation (*p*-value < 10^−20^). However, the changes in Ribo-seq and, importantly, the ribosome occupancy show a higher correlation (Pearson’s CC = 0.43 and 0.52, respectively) than the changes in RNA abundance (Pearson’s CC = 0.28). Thus, YB-1 and YB-3 likely substitute each other specifically at the level of translation. 

Interestingly, among the genes with significant changes in their ribosome occupancy, only about 30% of UP-regulated genes and about 10% of DOWN-regulated genes change in both YB-1- and YB-3-expressing cells (Figure 2b). Next, we compared changes in ribosome occupancy with YB-1- or YB-3 RNA binding specificity. We identified direct YB-1 (539 genes) and YB-3 (1600 genes) mRNA targets using PAR-CLIP (for YB-1, this study) and eCLIP (for YB-3, [12]), respectively. The YB-1 and YB-3 targets moderately change their ribosome occupancy upon YB-1 and YB-3 expression (Figure 2c, top panel). Interestingly, the YB-1 and YB-3 targets are enriched in DOWN-regulated genes in both YB-1- and YB-3-expressing cells (Figure 2c, bottom panel). This observation agrees with the published data regarding YB-1 and YB-3 as global translation inhibitors [10].

Thus, the moderate correlation of RO changes and a moderate RO decrease in direct YB protein mRNA targets allow for the conclusion that YB-1 and YB-3 can moderately inhibit the translation of their multiple mRNA targets. 

### 2.3. Translationally Altered Genes in YB-1- and YB-3-Expressing Cells Participate in Similar Biological Processes

To compare the functional groups of RNAs with altered ribosome occupancy, we applied the gene set enrichment analysis to the gene lists ranked by changes in ribosome occupancy. The analysis revealed several functional groups of genes that were similar for YB-1- and YB-3-expressing cells (Figure 3a). For example, UP-regulated mRNAs are related to RNA splicing and chromosome organization. DOWN-regulated mRNAs are related to oxidative phosphorylation and mitochondrial translation. Of note, the observed YB-1-specific functional groups are related to the processes shared between YB-1 and YB-3, including cytoplasmic translation (related to mitochondrial translation), mitochondrion organization (related to oxidative phosphorylation), and the cell cycle (related to chromosome organization). Thus, we conclude that in YB-1- and YB-3-expressing cells, mRNAs with an altered RO belong to closely related functional groups.

Next, in YB-1-/YB-3-expressing cells, we analyzed RO changes in YB-1- and YB-3-bound mRNAs (targets). For each functional gene group from the GSEA (Figure 3a), we checked the presence of YB-1 and YB-3 mRNA targets. As expected, the enrichment of YB-bound transcripts was exclusive to the down-regulated functional gene groups; these were then used for further analysis. Generally, for YB-1 and YB-3 targets, RO decreases more strongly than for other mRNAs (Figure 3b). The same effect was observed in YB-1- and YB-3-expressing cells for both YB-1 and YB-3 mRNA targets. Moreover, in YB-1-specific groups (cytoplasmic translation and mitochondrion organization) of the YB-3-expressing cells, despite non-significant RO changes in all mRNAs of this group, the mRNA targets have a significantly reduced RO. Thus, YB-1 and YB-3 can functionally replace each other in the inhibition of translation of their mRNA targets, which encode proteins involved in cytoplasmic and mitochondrial translation, oxidative phosphorylation, and mitochondrion organization.

## 3. Discussion

Here, we demonstrate that YB proteins have a dual effect on global translation. Compared to ΔΔ cells, the YB-expressing cells show an elevated level of translation. On the other hand, overexpression of the YB proteins causes the inhibition of global translation [10]. Moreover, the translation of direct mRNA targets identified by PAR and eCLIP was inhibited by the expression of YB proteins. YB-1 is known to both inhibit and stimulate the translation (for details, see ref. [1]) of specific mRNAs. Also, it was proposed that YB-1 can stimulate global translation by displacing initiation factors from the mRNA’s “body” and concentrating them at the 5′UTR [13]. However, the addition of YB-1 to a cell-free translation system based on either ΔΔ cells or ΔYB-1 cells does not stimulate the translation of reporter mRNAs [11]. This suggests that the main reason for the enhanced translation observed in YB-expressing cells is different from that proposed previously.

To explain the “translational duality paradox”, we analyzed changes in the ribosome footprint counts (with the Ribo counts reflecting the transcript-level abundance of translating ribosomes) of translation-related genes. The Ribo counts better reflect the protein amount than the RNA abundance or ribosome occupancy (translational efficiency); only a few genes changed by more than 40% in YB protein-expressing cells compared to the ΔΔ cells (Appendix A). Among these, only three are candidates for the regulation of global translation, namely *EIF4EBP1*, *EIF4EBP2*, and *SESN2*. Sestrin 2 is a known inhibitor of the mTOR signaling pathway [14,15], the master regulator of translation. 4EBP1/2 inhibit translation by sequestrating the cap-binding protein eIF4E [16,17]. Judging from the Ribo counts, upon YB protein expression, the protein abundance of 4EBP1 and Sestrin 2 decreases by 40%. We propose that the decreased expression of these proteins, in turn, causes the upregulation of the overall translation level.

Despite the two-fold increase of Ribo counts for *EIF4EBP2*, we do not expect any functional consequences because this 4EBP isoform is translated with a 10x lower efficiency in HEK293T compared to *EIF4EBP1* (Appendix A). Interestingly, according to PAR-CLIP and eCLIP, *IF4EBP1* mRNA has binding sites on YB-1 and YB-3 (Appendix A), probably as their direct regulatory target. 

The altered ribosome biogenesis might be another reason for the elevated translation observed in YB protein-expressing cells. In ΔΔ cells, the sucrose gradient peaks corresponding to 80S and 60S are higher than those for WT cells, but the 40S peaks are of the same height (Figure 1b). This can result from disproportion or defects in subunit maturation in ΔΔ cells. Interestingly, about 20 snoRNAs highly (more than 2-fold) changed at the level of Ribo counts but not RNA counts (Appendix A). It is known that Ribo-seq is contaminated with large non-ribosomal RNA–protein complexes, including those containing snoRNAs, responsible for rRNA modification [18]. We speculate that altered Ribo counts for snoRNAs reflect changes in the composition of rRNA modification complexes and ribosome subunit maturation. For example, the most increased association is observed for *SNORA65*, which pseudouridylates 28S rRNA around the peptidyl transferase center [19]; for *SNORD68*, the late-associating with pre-ribosome complexes snoRNA [20] that is responsible for the 2′-O-methylation of A2388 (28S) and U428 (18S) [21]; and for *SNORD27*, the long-associating snoRNA [20]. These observations support the hypothesis that YB protein expression in ΔΔ cells enhances correct ribosome maturation, which, in turn, increases the level of global translation. 

This study is focused on the functional interchangeability of YB-1 and YB-3. This phenomenon was previously demonstrated indirectly in experiments with knockout mice [22,23], where the double knockout YB-1^−/−^YB-3^−/−^ embryos had more severe malformations and died earlier (E8.5 to E11.5) compared to YB-1-deficient embryos (E18.5 to postnatal day 1) [23]. Thus, it was assumed that YB-3 could functionally replace YB-1 during the early stages of embryonic development. Lately, using RIP-seq, we found that YB-1 and YB-3 bind similar groups of mRNAs [10]. However, RIP-seq is known to detect both direct and indirect (mediated by protein partners) RNA targets. Here, using PAR-CLIP and eCLIP to primarily identify direct RNA targets, we demonstrate that the translation of YB-1 and YB-3 mRNA targets is inhibited upon both YB-1 and YB-3 expression. Though this translation inhibition level is relatively low, with the overall elevation of translation considered, the actual translation inhibition level relative to the mean is significantly higher. Moreover, the translationally changed genes in cells expressing YB-1 or YB-3 are related to similar cellular processes. Thus, we conclude that in HEK293T cells, YB-1 and YB-3 can substitute each other in their ability to inhibit the translation of their target mRNAs that encode proteins involved in translation, oxidative phosphorylation, and mitochondrion organization. 

## 4. Materials and Methods

### 4.1. Plasmids

The plasmids pcDNA3.1-puro-YB-1 and pcDNA3.1-puro-YB-3 were described previously [10]. Plasmid pcDNA3.1-puro-YB-1-YB-3 was obtained by ligating a fragment of the plasmid pcDNA3.1-puro-YB-3 (which contains the CMV promoter, YB-3 cDNA, and polyadenylation signal) cut from the indicated plasmid using *BglII* and *DraI* (blunted with T4 DNA polymerase), with plasmid pcDNA3.1-puro-YB-1 treated with the restriction endonucleases *BglII* and *NruI*.

### 4.2. Cell Cultures

The cells were cultivated in Dulbecco’s Modified Eagle’s Medium (DMEM) supplemented with 10% fetal calf serum, 2 mM glutamine, 100 U/mL penicillin, and 100 µg/mL streptomycin. The cells were incubated at 37 °C in a humidified atmosphere containing 5% CO_2_ and passaged by standard methods.

HEK293T cells (originally obtained from ATCC) were kindly provided by Dr. Elena Nadezhdina (Institute of Protein Research, RAS). HEK293TΔYB-1ΔYB-3 (ΔΔ) cells were generated from HEK293TΔYB-1 cells [10] using the CRISPR/Cas9n genome editing system [24]. The pSpCas9n(BB) -2A-Puro plasmids with guide RNAs (complementary to the first exon of the *YBX3* 5′-CAGGCTCCGACGGAGGCGGC-3′ and 5′-CCCCGCGCCCAAGAGCCCGG-3′) were linearized *BslI* and transfected into HEK293TΔYB-1 cells using Lipofectamine 3000 (Invitrogen, USA). After the treatment with puromycin (1 μg/mL) for three days, the cells were allowed to recover for two days, and then the pool of surviving cells was cloned by cell culture dilution and screened by immunoblotting.

The HEK293TΔYB-1ΔYB-3 + YB-1 (ΔΔ + YB-1) cell line was derived from the HEK293TΔYB-1ΔYB-3 cell line. To achieve the stable expression of YB-1, HEK293TΔYB-1ΔYB-3 cells were transfected with the *BglII*-linearized plasmid pcDNA3.1-Puro-YB-1. Twenty-four hours after the transfection, the cells were transferred into a medium with puromycin (the resistance to which was provided by the plasmid) and cultured for two weeks with the occasional replacement of the medium. Cultivation was then continued for another 2 weeks without puromycin, and YB-1 expression was confirmed by immunoblotting.

In a similar way, HEK293TΔYB-1ΔYB-3 + YB-3 and HEK293TΔYB-1ΔYB-3 + YB-1 + YB-3 cells were obtained by transfecting HEK293TΔYB-1ΔYB-3 cells with linearized *BglII* pcDNA3.1-puro-YB-3 or pcDNA3.1-puro-YB-1-YB-3 plasmids, respectively.

### 4.3. Western Blot and Antibodies

Cell lysates (see Section 4.5) were separated by SDS–PAGE and transferred onto a nitrocellulose membrane. The membrane was blocked for 1 h at room temperature with non-fat 5% milk in TBS (10 mM Tris-HCl, pH 7.6, and 150 mM NaCl) and incubated overnight at 4 °C in TBS-T (10 mM Tris-HCl, pH 7.6, 150 mM NaCl, and 0.05% Tween 20) supplemented with BSA (7.5%) and the appropriate antibodies. The membrane was washed three times with TBS-T, incubated for 1 h with 5% non-fat milk in TBS-T, and secondary antibodies conjugated with horseradish peroxidase, and then washed three times with TBS-T. Immunocomplexes were detected using an ECL Prime kit (GE Healthcare, Chicago, IL, USA) according to the manufacturer’s recommendations. Polyclonal rabbit antibodies against YB-1 (Y0396, Sigma-Aldrich, Burlington, MA, USA), polyclonal rabbit antibodies against YB-3 (A303-070A, Bethyl, Montgomery, TX, USA), polyclonal rabbit antibodies against RPL7 (SAB4502656, Sigma-Aldrich), polyclonal rabbit antibodies against RPS6 (2217, Cell Signaling, Danvers, MA, USA), and anti-rabbit HRP-conjugated antibodies (7074, Cell Signaling) were used.

### 4.4. Sucrose Gradient 

The cells were washed twice with ice-cold PBS containing 0.1 mg/mL cycloheximide and lysed directly on the plate after the addition of 400 µL of polysome extraction buffer: 15 mM Hepes-KOH, pH 7.6, 5 mM MgCl_2_, 0.3 M NaCl, 1% Triton X-100, 0.1 mg/mL cycloheximide, 0.2 mM VRC (vanadylribonucleoside complex), and 1 mM DTT. The extracts were transferred into 1.5-mL tubes and incubated on ice for 10 min with the occasional mixing. The nuclei and debris were removed by centrifugation at 12,000× *g* for 10 min in a microcentrifuge. Forty-five-µL aliquots of supernatants were layered onto 4.5 mL of a 15–45% sucrose gradient in buffer (15 mM Hepes-KOH, pH 7.6, 5 mM MgCl_2_, 100 mM KCl, 0.1 mM EDTA, and 0.01 mg/mL cycloheximide), and centrifuged in a SW-60 rotor (Beckman Coulter, Brea, CA, USA) at 45,000 rpm for 55 min at 4 °C. All gradients were monitored for absorbance at 254 nm during their collection from the bottom. A total of 0.375-mL fractions were collected and combined into larger ones: HPs—heavy polysomes (fractions 1–6), LPs—light polysomes (fractions 7–9), MonoDi—monosomes and disomes (fractions 10–12), and free—free mRNPs (fractions 13–16). A total of 0.1 ng of in vitro-transcribed *Nanoluc luciferase* (*Nluc*) mRNA [10] was added to each fraction for normalization. 

### 4.5. Ribo-Seq and RNA-Seq

HEK293TΔYB-1ΔYB-3, HEK293TΔYB-1ΔYB-3 + YB-1, and HEK293TΔYB-1ΔYB-3 + YB-3 cells at 70–80% confluency were immediately chilled on ice and washed with PBS + cycloheximide (100 µg/mL). The cells were not pre-treated with cycloheximide to avoid the artificial accumulation of initiation complexes at the translation initiation stage [25]. The cells were then lysed with buffer containing 20 mM Tris-HCl (pH 7.4), 150 mM NaCl, 5 mM MgCl_2_, 1 mM DTT, 1% Triton X-100, 100 µg/mL cycloheximide (Sigma-Aldrich, Burlington, MA, USA), and 25 U/mL TURBO DNase (Ambion, Invitrogen, Thermo Fisher Scientific, Waltham, MA, USA). Cell lysates were incubated on ice for 10 min, triturated ten times through a 26-G needle, and centrifuged at 20,000× *g* at +4 °C for 10 min. The supernatant was divided into two parts for Ribo-seq and RNA-seq library preparation. Nuclease footprinting and ribosome recovery for Ribo-seq library preparation were performed according to [26]. The adapter for Ribo-seq (5′-/5Phos/NNNNNATCGTAGATCGGAAGAGCACACGTCTGAA/3ddC/-3′) had an UMI sequence (5 random nucleotides) to allow for deduplication. Total RNA for RNA-seq library preparation was isolated using TRIzol LS Reagent (Thermo Fisher Scientific, Waltham, MA, USA). rRNA depletion was performed using a RiboMinus™ Eukaryote Kit v2 (Thermo Fisher Scientific, Waltham, MA, USA) according to the manufacturer’s recommendations. RNA-seq libraries for sequencing were obtained using a NEBNext Ultra Directional RNA Library Prep Kit for Illumina (NEB, Ipswich, MA, USA) according to the manufacturer’s protocol. 

### 4.6. PAR-CLIP 

HEK293T cells were grown to a density of about 80%, treated with 4-thiouridine (100 uM) for 16 h, and irradiated in a Bio-link BLX-365 UV irradiation system (Vilber, Collégien, France) 0.15 J/cm^2^ (wavelength: 365 nm). Then, the cells were washed twice with cold PBS, pelleted by centrifugation (500 g), and lysed with 3 volumes of NP40 buffer (50 mM Hepes-KOH (pH 7.5), 150 mM KCl, 2 mM EDTA, 0.5% NP40, 0.5 mM DTT, and a mixture of protease inhibitors (Roche, Basel, Switzerland)), and the cell debris was pelleted at 13,000× *g* for 10 min at 4 °C. 

Further steps of PAR-CLIP were performed according to Danan et al. [27]. For immunoprecipitation, protein G-Sepharose (GE Healthcare, Chicago, IL, USA) was mixed in a 1:5 volume ratio with 5% BSA prepared with NT2 buffer (50 mM Tris-HCl (pH 7.6), 150 mM NaCl, 1 mM MgCl_2_, and 0.05% NP40). 300 μL of protein G-Sepharose suspension per immunoprecipitation reaction was incubated overnight at 4 °C with antibodies against YB-1 (10 μg). The resin was washed with cold NT2 buffer and resuspended in 900 μL of NT2 buffer supplemented with 1 mM DTT and 20 mM EDTA. The lysate was treated with RNase T1 (1 unit/μL) at 22 °C for 15 min, added to the resin, and incubated on a rotary mixer at 4 °C for 1 h. Resin was washed twice with NP40 buffer and then additionally incubated with RNase T1 (1 unit/µL) in NP40 buffer at 22 °C for 15 min. Then, the resin was washed twice with NP40 buffer, twice with dephosphorylation buffer (50 mM Tris-HCl (pH 7.9), 100 mM NaCl, 10 mM MgCl_2_, and 1 mM DTT), and resuspended in 400 μL of dephosphorylation buffer. Next, alkaline phosphatase CIAP (Thermo Fisher Scientific, Waltham, MA, USA) was added to 0.5 units/μL, and the mixture was incubated at 37 °C for 10 min. Resin was washed twice with NP40 buffer and once with buffer for T4 polynucleotide kinase without DTT (50 mM Tris-HCl (pH 7.5), 50 mM NaCl, 10 mM MgCl_2_), and resuspended in T4 polynucleotide kinase buffer with 5 mM DTT. T4-polynucleotide kinase (1 unit/μL) and [γ-^32^P]-ATP (0.5 μCi/μL) were added, and the mixture was incubated at 37 °C for 30 min to label RNA fragments cross-linked with the protein (YB-1), and then another 5 min in the presence of 100 μM ATP to phosphorylate the 5′ ends of all RNA fragments not phosphorylated in the presence of [γ-^32^P]-ATP. Next, the resin was washed five times with buffer for T4 polynucleotide kinase without DTT, and proteins bound to the resin were eluted with 30 μL of acid–urea buffer. Proteins were separated by electrophoresis in an acid-urea 10% polyacrylamide gel [28], and the YB-1 protein labeled with an RNA fragment was detected by autoradiography. The gel fragment corresponding to YB-1 was excised, and the protein was electroeluted from the gel using D-Tube Dialysers Midi (Novagen, Merck, Darmstadt, Germany) in 400 μL of SDS gel electrophoresis buffer. After electroelution, an equal volume of 2X proteinase K buffer (100 mM Tris-HCl (pH 7.5), 150 mM NaCl, 12.5 mM EDTA, and 2% SDS), and proteinase K (1.2 mg/mL) were added to the sample and incubated at 55 °C for 30 min. Next, RNA was isolated from the mixture using phenol deproteinization, and the RNA was reprecipitated with ethanol. The RNA pellet was resuspended in 5 μL of water. At the next stage, based on the isolated RNA fragments, a library was prepared for high-throughput sequencing using an ARTseq™ Ribosome Profiling Kit (Epicentre, Illumina, San Diego, CA, USA), starting with the stage of ligation of the 3′ adapter and then following the protocol. 

### 4.7. High-Throughput Sequencing and Data Processing

The libraries (Appendix A) were sequenced on Illumina NextSeq 500 and HiSeq 2000. Read quality control was performed with FastQC v0.11.5 [29]. The reads were trimmed with cutadapt v2.10 [29], removing short reads (<20 nt) and adapter sequences. Deduplication was performed using a seqkit (v2.6.1) [30]. Read mapping was performed to the hg38 genome assembly with the GENCODE v29 basic genome annotation using STAR v2.7.6a [31]. The resulting read counts from several sequencing runs of the same sample were summed up. The principal component analysis of the samples shows a reasonable separation of experiment types and agreement between replicates (Appendix A). The Ribo-seq data have a clear triplet periodicity. Metagene profiles of windows surrounding gene start codons were obtained with plastid v.0.6.1 [32] and are visualized in Appendix A. Gene counts for rRNA and mt-rRNA were excluded.

The differential gene expression and gene set enrichment analyses were performed in the R environment. The *ComBat-seq* function from sva (v.3.35.2) [33] was used for batch correction (the Ribo-seq and RNA-seq samples were corrected independently), with edgeR (v.3.28.1) [34] used to identify differentially expressed genes among 10832 genes that passed 2 cpm (default TMM normalization with *calcNormFactors*, dispersion estimation with estimateDisp, and test for differential expression with *exactTest*). Differential gene expression (changes in RNA-seq and Ribo-seq) and ribosome occupancy (Ribo-seq relative to RNA-seq, RO) were estimated using contrasts of the edgeR generalized linear model (*glmQLFit* and *glmQLFTest*). *p*-values were corrected for multiple testing using the FDR (Benjamini–Hochberg) procedure. Verification of the Ribo-seq and RNA-seq data is presented in Appendix A.

The gene set enrichment analysis (GSEA) was performed using a fgsea (v.1.24.0) package [35]. GO terms were obtained from MSIGdb 3.0 [36]. Gene ranking was obtained using the signed adjusted *p*-value (sign taken from the fold change, more significant up- and down-regulated genes appear at the top and bottom of the list, respectively). Clustering of significantly enriched (FDR < 0.05) GO terms from GSEA analysis (biological processes, BP, annotation) was performed using the *GO_similarity* and *cluster_terms* functions from a simplify Enrichment (v.1.9.5) package [37]. 

### 4.8. PAR-CLIP and eCLIP Data Processing

The YB1 PAR-CLIP processing pipeline was based on [38]. Trimming was performed with cutadapt *-a AGATCGGAAGAGCACACGTCTGAACTCCAGTCACNNNNNNATC TCGTATGCCGTCTTCTGCTTG -m 15 -q 20*. Next, the trimmed reads were aligned to the human hg38 genome with bowtie (v.1.0.0) [39] with the following parameters: *--best --chunkmbs 512 -n 1 -S -M 100*. The resulting alignment files of two independent replicates (Rep.A and Rep.B) were supplied to the wavClusteR package (v.2.34.0) [40] to perform peak calling. For that, only the reads mapped to the reference chromosomes were obtained with samtools (v.1.13) [41] *view*, followed by the *readSortedBam* function from wavClusteR. Next, *getAllSub* with *minCov = 3* and *fitMixtureModel* with *substitution = “TC”* were used for the read bam files in order to obtain the substitution regions and discriminate PAR-CLIP-specific and extrinsic transitions. Finally, *getHighConfSub*, *getClusters* with *threshold = 1*, and *filterClusters* with *refBase = “T”* and *minWidth = 10* were used to identify the PAR-CLIP peaks that were exported with *exportClusters*. Overall, we identified 1401 and 2042 peaks for Rep.A and Rep.B, respectively. To annotate the PAR-CLIP peaks, they were intersected with the RefSeq annotation downloaded from the UCSC table browser using bedtools (v.2.30.0) [42] *intersect -wa -wb -s*. For further analyses, we only used the peaks reproducible between replicates. 

To identify the eCLIP RNA targets of YB-3 in K562 and HepG2 cells, we used ENCODE eCLIP and followed the protocol as in [43]. The reads were preprocessed in the same way as in [12], including adapter trimming with cutadapt (v.1.18) [29]. The preprocessed reads were mapped to the hg38 genome assembly with GENCODE v31 comprehensive annotation using hisat2 (v.2.1.0) [44] and deduplicated using the barcodecollapsepe.py script from [45]. Properly paired and uniquely mapped second reads were extracted using samtools (v.1.9, with *-f 131 -q 60*) [46]. Gene-level read counts were obtained with plastid (v.0.4.8) [32] by counting the 5′ ends of the reads. The analysis of specific enrichment against size-matched control experiments was performed with edgeR (v.3.18.1) [34], considering only genes passing 2 counts per million in at least 2 of the 3 available replicates. The RNA targets were defined at 5% FDR and log_2_(fold change) > 0.5.

## Figures and Tables

**Figure 1 ijms-25-01736-f001:**
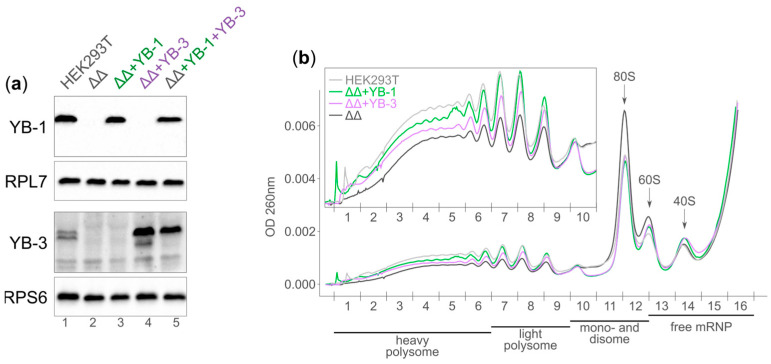
YB-1 and YB-3 expression partially restores translation in HEK293ΔYB-1ΔYB-3 cells. (**a**) Western blot analysis of lysates of HEK293T, HEK293TΔYB-1ΔYB-3 (ΔΔ), and ΔΔ cells expressing YB-1 and/or YB-3. RPS6 and RPL7 were used as loading controls. (**b**), The polysome profiles of HEK293T (light gray), HEK293TΔΔ (dark gray), HEK293TΔΔ + YB-1 (green), and HEK293TΔΔ + YB-3 cells (purple). Peaks corresponding to 80S, 60S, and 40S are marked.

**Figure 2 ijms-25-01736-f002:**
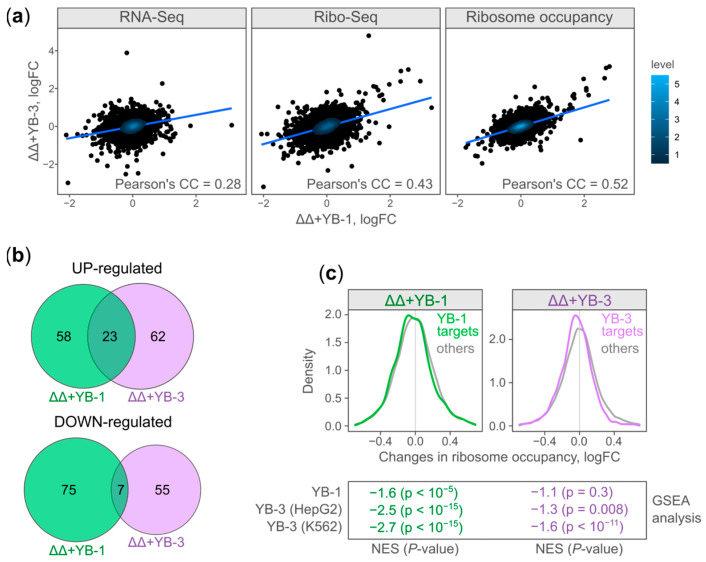
Changes in ribosome occupancy upon YB-1/YB-3 expression in HEK293TΔYB-1ΔYB-3 cells. (**a**) Scatterplots illustrating the changes in RNA abundance (RNA-seq, left panel), ribosome footprint counts (Ribo-seq, middle panel), and ribosome occupancy (RO, right panel) in YB-1-expressing cells (X-axis) plotted against the respective changes in YB-3-expressing cells (Y-axis), both cell lines compared to ΔΔ. The two-dimensional kernel density estimation, linear regression line, and Pearson’s correlation coefficient are shown. (**b**) A Venn diagram illustrating the number of shared and unique significantly UP- (RO_logFC > 0.5, FDR < 0.05) and DOWN- (RO_logFC < −0.5, FDR < 0.05) regulated genes in YB-1/YB-3-expressing cells. (**c**) Density plot of RO changes in YB-1- (left panel) and YB-3- (right panel)expressing cells. YB-1- or YB-3-bound mRNAs (mRNA targets) identified by PAR-CLIP (photoactivatable ribonucleoside-enhanced crosslinking and immunoprecipitation) or eCLIP (enhanced crosslinking and immunoprecipitation), respectively. Bottom panel: mRNAs with a decreased RO are enriched with YB-1- and YB-3-bound mRNAs. GSEA: gene set enrichment analysis (GSEA).

**Figure 3 ijms-25-01736-f003:**
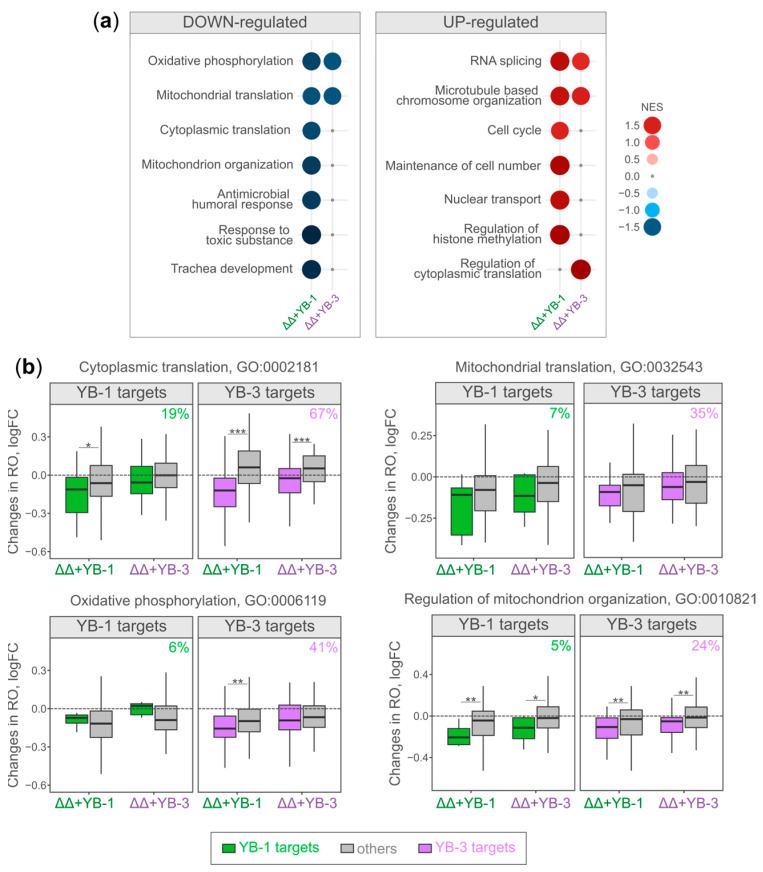
Biological processes involving mRNAs with a changed ribosome occupancy in YB-1-/YB-3-expressing cells. (**a**) The balloon plot of GO terms clustered by similarity. Only significantly enriched (FDR < 0.05) GO terms from the GSEA analysis (biological processes, BP, annotation) were used for clustering. Mean normalized enrichment scores (NESs) of GO terms in the clusters are shown. (**b**) The boxplots show RO changes in non-target (gray), YB-1 (green, left panels), or YB-3 (purple, right panels) mRNA targets related to the indicated GO term in YB-1-/YB-3-expressing cells. Center lines show the medians; box limits indicate the 25th and 75th percentiles; whiskers denote the 1.5x interquartile range; and the outliers are hidden. The percentage of YB-1 (green) or YB-3 (purple) mRNA targets from all mRNAs in the GO term is shown. The significance of RO changes between target and non-target mRNAs is estimated with the one-sided Mann–Whitney test. * *p* < 0.1, ** *p* < 0.05, and *** *p* < 0.01.

## Data Availability

The data are deposited to the NCBI GEO under the accession number GSE249897.

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
