# Peer review of "Y-Box-Binding Proteins Have a Dual Impact on Cellular Translation"

_ijms, 2024, doi:10.3390/ijms25031736_

Round 1
Reviewer 1 Report
Comments and Suggestions for Authors
Eliseeva et al.'s manuscript focuses on the dual impact of YB-1 and YB-3 proteins on cellular translation. Based on a series of experiments with various knockout variants of HEK293T cells, the authors concluded that two YB proteins could functionally replace each other during the inhibition of translation of appropriate mRNAs. The authors put effort into this work, and the experiments were performed in a detailed manner. The paper is well-written, and the results obtained support conclusions. Therefore, I suggest to accept the manuscript to be published in the International Journal of Molecular Science.
Author Response
Thank you for the high evaluation of our work.
Reviewer 2 Report
Comments and Suggestions for Authors
This study explores the functional interchangeability of somatic YB-proteins, YB-1 and YB-3, in the regulation of translation. With HEK293T cells, the researchers employed RNA-Seq and Ribo-Seq techniques to analyze changes in mRNA abundance and mRNA translation in cells expressing only YB-1, only YB-3, or neither of them. The authors suggest that YB-1 and YB-3 can substitute for each other in translation regulation. While the study explores the functional interchangeability of YB-1 and YB-3 in translation regulation, there are several major issues.
Experimental Design and Clarity:
The manuscript lacks clarity regarding the experimental design and execution. It is essential to provide detailed information on how the experiments were conducted, including the number of replicates, controls, and any potential technical biases. The methods section should be expanded to describe the procedures, protocols, and instrumentation used in sufficient detail to allow for replication by other researchers. The lack of clarity in the experimental design hinders the reproducibility and transparency of the study.
Data Accessibility:
The data generated in this study must be made publicly available to ensure transparency and reproducibility. While the manuscript mentions data deposition in NCBI GEO under accession number GSE249897, it is noted that the data is currently private and will only be released in December 2025. This extended embargo period raises concerns about the accessibility of the data for the scientific community. Authors should consider making the data available as soon as possible to facilitate further analysis and validation by other researchers.
Conclusion Validity:
The conclusion that YB-1 and YB-3 can functionally replace each other in translation regulation appears premature and lacks comprehensive support from the study design. To establish this conclusion, the study should incorporate a more thorough investigation of the functional differences and similarities between YB-1 and YB-3. This may involve a more extensive set of experiments, and additional functional assays. Without a more comprehensive study design, the conclusion remains speculative.
Author Response
We are very grateful for the thorough reviewing of our manuscript. Thank you for valuable comments and suggestions. We did our best to address these issues in the revised version of the manuscript.
Experimental Design and Clarity:
The manuscript lacks clarity regarding the experimental design and execution. It is essential to provide detailed information on how the experiments were conducted, including the number of replicates, controls, and any potential technical biases. The methods section should be expanded to describe the procedures, protocols, and instrumentation used in sufficient detail to allow for replication by other researchers. The lack of clarity in the experimental design hinders the reproducibility and transparency of the study.
We did our best to expand the Methods section by including more details regarding the experimental procedure, particularly, covering PAR-CLIP and Ribo-Seq protocol (sections 4.5 and 4.6) and eCLIP data processing (section 4.8). Further, we have added an extensive supplementary table (Table S2, Supplementary materials) that provides the description of samples, replicates and controls, sequencing and library preparation batches, sequencing instruments used, and the total read counts.
Data Accessibility:
The data generated in this study must be made publicly available to ensure transparency and reproducibility. While the manuscript mentions data deposition in NCBI GEO under accession number GSE249897, it is noted that the data is currently private and will only be released in December 2025. This extended embargo period raises concerns about the accessibility of the data for the scientific community. Authors should consider making the data available as soon as possible to facilitate further analysis and validation by other researchers.
Indeed, the original embargo period was clearly excessive. We have lifted embargo on the respective GEO SuperSeries (GSE249897) and SubSeries (GSE249895, GSE249896) are now public.
Conclusion Validity:
The conclusion that YB-1 and YB-3 can functionally replace each other in translation regulation appears premature and lacks comprehensive support from the study design. To establish this conclusion, the study should incorporate a more thorough investigation of the functional differences and similarities between YB-1 and YB-3. This may involve a more extensive set of experiments, and additional functional assays. Without a more comprehensive study design, the conclusion remains speculative.
Indeed, the roles of YB proteins in translation control are diverse and context-specific, and our original conclusion could be rephrased in a more careful fashion. Further, our study is not the first one focused on YB-1 and YB-3 functional interchangeability. YB-1 and YB-3 homology, their roles in RNA-dependent processes, and the potential for mutual functional replacement was actively discussed in the literature (ref. 1;2;10;23 in the manuscript). The first indirect support was obtained in mice knockout experiments (see Discussion and ref. 23). However, at that moment, the nature of the processes in which YB-1 and YB-3 can replace each other remained unexplored. In our previous publication (ref. 10 in the manuscript), we demonstrated that YB-1 knockout leads to YB-3 overexpression. Moreover, YB-1 and YB-3 bind similar sets of RNAs and, in the absence of YB-1, YB-3 enhances its mRNA binding and, potentially, could functionally replace YB-1 in RNA-dependent processes (see Introduction and ref. 10). The study described in our manuscript is the next step towards revealing details of the YB-proteins functional interchangeability. Here, we used double knockout (YBX1 and YBX3) cells and cells with restored expression of YB-1 or YB-3. To assess the global impact of YB-proteins on translation we used two different approaches: gradient profiles (Fig. 1b, section 2.1) and azidohomoalanin incorporation into newly synthesized proteins (Fig. S1, section 2.1); we have highlighted the fact of two independent different experimental approaches in the revised manuscript (section 2.1).
Next, to reveal more details of YB-protein involvement in translation, we performed ribosomal profiling (Ribo-Seq) and found that the translational changes upon YB-1 or YB-3 are indeed similar. We additionally analyzed the translation of direct mRNA targets determined by PAR-CLIP (for YB-1, described in this study) and eCLIP (for YB-3) and found that both YB-1 or YB-3 inhibit translation of both YB-1 and YB-3 mRNA targets.
Thus, our conclusion of YB-1 and YB-3 functional interchangeability is based on multiple different observations discovered in the presented manuscript as well as in previous studies. We did our best to convey this overall message in the revised version of the manuscript.
There remains an open possibility for performing different assays to identify other proteins and RNAs involved in the molecular mechanism of YB-protein-dependent translational control, but it does not seem to fit the scope of the presented manuscript.
Round 2
Reviewer 2 Report
Comments and Suggestions for Authors
The revised paper shows improvement. However, the conclusion stating that "YB-1 and YB-3 can functionally replace each other in translation regulation" remains overly generalized. It is essential to provide specificity regarding the cell type and contextual circumstances under which this functional replacement occurs. Further clarification on these aspects would enhance the accuracy and applicability of the conclusion.
Author Response
Thank you for clarifying your suggestion. We did our best to clarify the statement by specifying the cell type and the particular functional groups to which regulated mRNA targets are related (see lines 17-22, 48-53, 157-159,230-234).